# Lead yourself to the zone and be happy: The effect of self-leadership development on flow and happiness

Ricardo J. Vargas[1,2]*, Osvaldo Santos[3,4], Mónica Fialho[3], Joana Costa[5], Nicole Eifler[2], Pedro Marques-Quinteiro[6,7], Luís Curral[1,8]

1 Faculdade de Psicologia, Universidade de Lisboa, Lisboa, Portugal, 2 Consulting House, Lisboa, Portugal, 3 Faculdade de Medicina, Environmental Health Behaviour Lab, Instituto de Saúde Ambiental, Universidade de Lisboa, Lisboa, Portugal, 4 Unbreakable Idea Research, Lisboa, Portugal, 5 LEAF - Linking Landscape, Environment, Agriculture and Food Research Centre, Instituto Superior de Agronomia, Universidade de Lisboa, Lisboa, Portugal, 6 Intrepid Lab, ECEO, Lusofona University, Lisbon, Portugal, 7 CETRAD, Centro de Estudos Transdisciplinares para o Desenvolvimento, UTAD, Trás-os-Montes e Alto Douro, Portugal, 8 Research Center for Psychological Science – CICPSI, Universidade de Lisboa, Lisboa, Portugal

* ricardo.vargas@consulting-house.eu

## Abstract

### Background

Self-leadership has been associated with several positive organizational outcomes (e.g., performance and innovation). Yet, individual subjective well-being constructs have seldom been explored in the self-leadership literature. We hypothesized that an increase in self-leadership results in increased positive affect resources, namely flow and happiness.

### Methods

A self-leadership intervention program, interlinked with a real-world longitudinal observational effectiveness-assessment design, was used to test this hypothesis. A sample of 64 middle-managers from a for-profit organization in the fast-moving consumer goods (FMCG) sector went through one 8-week self-leadership training program, and provided 4383 daily measures of self-leadership, flow, and happiness during working hours on business days, plus 242 post-training modules assessments. Email surveys were used to assess self-leadership and dispositional flow, whereas the experiencing sampling method was used to assess situational (i.e., at-the-moment) flow and happiness. Linear mixed models and mediation analysis were applied to longitudinal data.

### Results

As hypothesized, an increase in self-leadership was positively associated to (1) situational and dispositional experiences of flow among employees in the FMCG

**Data availability statement:** All collected data are available, together with the R syntax needed to reproduce the results, on the Zenodo platform (https://doi.org/10.5281/zenodo.15395468).

**Funding:** The second, third, and sixth authors (OS; MF; PMQ) received national funding from FCT – Fundação para a Ciência e a Tecnologia (UIDB/04295/2020; UIDP/04295/2020; UID/04011). The funder did not play any role in the study design, data collection and analysis, decision to publish, nor preparation of the manuscrip. URL of funder: https://www.fct.pt/en/.

**Competing interests:** I have read the journal's policy and the authors of this manuscript have the following competing interests: the first and fifth authors are consultants with the organization that delivered the self-leadership training reported in the current paper. This does not alter our adherence to PLOS ONE policies on sharing data and materials.

sector and (2) happiness in the workplace. No mediation of flow was found between self-leadership and happiness.

## Conclusion

Results highlight the potential of self-leadership skills development and practice to shorten the distance between perceived challenges and skills in job-related tasks, as well as to make subjects feel happier in the workplace.

## Introduction

Business organizations must attract and retain talent to be competitive. Companies with happy employees are perceived as good places to work and thus can create and maintain competitive teams in the short and long term [1,2]. Besides different organizational cultures and contextual events that impact each job differently, employee happiness also depends on individual characteristics, namely personality, attitudes and beliefs, skills, and motivations [3]. Furthermore, individual characteristics evolve throughout lifespan, and much can be done by organizations to promote employees' subjective well-being (e.g., training; work-life balance; coaching). There is nevertheless a gap in knowledge about how companies can equip their employees with the most adequate individual competencies to deal with daily job demands in ways that promote employees' abilities to positively influence their own levels of happiness at work.

Over the last four decades, self-leadership has gained momentum in organizational psychology [4–7], being proposed as a self-influence process "that concerns leading oneself toward performance of naturally motivating tasks as well as managing oneself to do work that must be done but is not naturally motivating" [8]. Self-leadership may contribute to the development of self-regulatory strategies in job-related tasks [7,8], and empower individuals to thrive under rapidly changing and unpredictable work environments [8–10]. Indeed, some evidence supports the idea that individuals who self-regulate under stressful and unpredictable working conditions (i.e., job demands) are more likely to perform adaptively and achieve higher levels of job satisfaction [11–13], with positive outcomes for the individuals themselves and the organizations [5,6]. Although related concepts, work-related happiness is a broader concept, including job satisfaction but going beyond it to encompass positive affect and psychological functioning at work [14]. A main argument for the utility of self-leadership in organizational psychology relates to the idea that self-leadership provides work-related autonomy and, therefore, integrated self-regulation, through the implementation of psychological strategies, such as goal setting, self-cueing, self-rewards, among others [7,8,15].

The focus on promoting self-regulation and autonomy finds roots in the theory of self-determination [16], which is a major framework for understanding human motivation. According to this theory, autonomy (together with competence and relatedness) is a pivotal psychological need, crucial for subjective well-being and, synergically,

for organizational growth [17]. According to the self-determination theory, much of what people do at the workplace is not intrinsically motivating [16], autonomy brings with it the need for individuals to purposefully influence themselves, i.e., to adopt an effective self-leadership [4–6,18], which implies self-regulation of individuals' adaptive processes and skills for coping with work challenges.

## Self-leadership competencies as resources for coping with job demands

For clarification, self-leadership entails three components rooted in social cognitive theory [7]: behavior-focused strategies, constructive thought pattern strategies and natural reward strategies. Behavior-focused strategies refer to the self-monitoring of current actions and the decision on their suitability (implying self-observation, self-regulation, and self-reflection). Constructive thought pattern strategies make use of self-talk and mental imagery combined with the assessment of own values and beliefs to identify, and eventually replace, dysfunctional beliefs (based on symbolization and vicarious learning skills) and, ultimately, promoting perception of self-efficacy. Finally, natural rewards strategies refer to the restructuring and/or reframing of unpleasant tasks in the pursuit and promotion of pleasant experiences at work, implying forethought skills [8,13,15,19]. Combining these behavioral and cognitive strategies, it is possible to develop and practice self-leadership. And, when well-articulated, these strategies were found to be helpful in building autonomous motivation to influence positive affect and performance during job-related tasks (e.g., Breevaart et al. [11]; Manz [8]; Marques-Quinteiro et al. [12]), even when coping with high-level or unpredictable job demands. Indeed, self-leadership has been positively associated with adaptive performance in the workplace (e.g., Marques-Quinteiro et al. [12]; Neck and Manz [13]; Roberts and Foti [20]), career self-management and success (e.g., Murphy and Ensher [21]; Raabe, Frese and Beehr [22]), and creativity and innovation (e.g., Amundsen and Martinsen [23]; Ghosh [24]; Pratoom and Savatsomboon [25]). Consistently, a negative association between self-leadership and absenteeism (e.g., Frayne and Latham [26]), stress and anxiety [27–29] has been reported. Evidence also shows that individuals can train their self-leadership skills and increase their use of self-leadership-related psychological strategies (e.g., Furtner, Sachse and Exenberger [30]; Latham [31]; Marques-Quinteiro et al. [12]; Stewart et al. [7]) while maximizing the positive outcomes of adopting such strategies. Shortly, self-leadership is a competence that can be developed, strengthening the self-regulation process that is required for employees to thrive in their job careers.

The Job-Demands Resources (JD-R) model [32–36] has been used as a theoretical framework for understanding the organizational antecedents of self-leadership [37]. Although self-leadership development typically focuses on enhancing psychological (behavioral, cognitive and emotional) resources that contribute to meeting work-related goals, personal growth, learning, and development of the employee, together with other social and organizational job resources (counteracting against job demands, according to the JD-R model), to the best of our knowledge it remains unclear if subjective well-being (namely, happiness) is also an effect of developing self-leadership competencies. The integration of self-leadership and work-related happiness, using the framework of the JD-R, has not been considered in the literature and is therefore a theoretical contribution of this study.

## Self-leadership as a potential facilitator of work-related happiness

Happiness has been defined in multiple ways (see Diener, Scollon and Lucas [38]; Fisher [14]; Frawley [39]). There are two main perspectives about this concept: happiness as the outcome of pleasant feelings and judgments (i.e., hedonic view of happiness), and happiness as an outcome of acting according to own values and beliefs (i.e., eudaimonic view of happiness) [14,40]. Ruut Veenhoven pioneered the study of happiness and defined this construct as 'the overall appreciation of one's life as-a-whole' [41]. According to this line of thought, happiness is frequently referred to as a close entity of life satisfaction and of overall subjective well-being [42]. The same author considers two interlinked domains of happiness: the affective (*how well do I feel?*) and cognitive (*how far am I from my life expectations?*) domains [42]. In the workplace

environment, happiness has been assessed through various related constructs that capture pleasant judgments and experiences, the most common one being job satisfaction [14]. When considering the consequences of happiness for organizations, a distinction between the effects of momentary states of happiness (i.e., transient or situational happiness) and person-level happiness (i.e., dispositional happiness) is helpful and aligned with Veenhoven's dual-domain perspective. Empirical research has shown that situational happiness is positively associated with creativity and proactivity [43,44], and negatively associated with interpersonal conflict [45]. Regarding dispositional happiness, it has been associated with job satisfaction and organizational commitment, which are, in turn, negatively associated with absenteeism (e.g., Wegge, Schmidt, Parkes and van Dick [46]) intention to quit, and turnover [47,48] as well as with anxiety, depression and burnout (reviewed in Faragher, Cass and Cooper [49]). In sum, happiness in the workplace is beneficial for both employees and organizations [14].

It is again possible to advance an association between self-leadership, as a process linked with "naturally motivating tasks" [8] and its self-regulatory and autonomy-promoting nature, and happiness. As a matter of fact, self-leadership goes beyond the achievement of self-goals in a predefined sequence, including also the subjective positive experience of performing job-related tasks and, consequently, positive subjective well-being experiences [5]. Indeed, self-leaders take ownership, make choices (i.e., are autonomously regulated) and feel competent to execute their tasks, over the conditions and processes of job-related tasks, which ultimately may impact their subjective well-being (see Goldsby et al. [5]; Harari et al. [6]; Stewart et al. [7]).

At this point, it is important to highlight that according to the self-determination theory, the combination of feeling autonomous with feeling competent (i.e., perceiving self-efficacy for facing specific demands) is a major motivational driver to repeat non-intrinsically motivating behaviors [16]. Since self-leadership training targets both the perception of autonomy and competence, it is theoretically possible to argue that self-leadership increases motivation to endure in highly demanding jobs. An open question is how this self-regulation process can have an impact on happiness. Indeed, being confident in performing job-related tasks is surely not a sufficient condition, *per se,* to feel happy; therefore, one cannot assume without better evidence that self-leadership is a predictor of happiness.

Both increased performance and positive affect are expected after a self-leadership training [15]. Previous studies have found significant differences in positive affect between experimental (receiving self-leadership training) and control groups (e.g., Marques-Quinteiro et al. [12]; Neck and Manz [13]). As discussed in the introduction section, self-leadership is associated with self-regulation (which can be viewed as a proxy of self-perception of autonomy) [7,8], as well as with the perception of being equipped for facing job demands [8–10], which can be viewed as a proxy for self-perception of competence. Autonomy and competence are, according to the self-determination theory, two fundamental psychological needs that drive motivation to adhere to non-intrinsically motivating tasks (such as work-related tasks) and, ultimately, to achieve eudaimonia [16]. Therefore, taking into account the self-determination theory, self-leadership has the potential to promote the satisfaction of basic psychological needs and, therefore, the potential to increase employees' happiness. From here, we may formulate hypothesis 1.

**Hypothesis 1** Self-leadership variation across time is associated with happiness variation in the workplace.

## Flow as a potential effect of self-leadership

Flow has been defined 'as a subjective state that people report when they are completely involved in something to the point of forgetting time, fatigue, and everything else but the activity itself [50]. Bakker [51] applied this original flow construct to the workplace, referring to short-term peak job-related experiences during which the subject is intrinsically motivated for the tasks being performed and enjoys them, while experiencing an altered sense of time. This can be stated as an optimal experience and occurs whenever perceived challenges and skills are well-balanced, i.e., when one's skills match the demands to complete a task. When challenges and skills are mismatched, the subject might become anxious (high demands versus low skills) or bored (low demands versus high skills) [52–54]. In flow episodes, besides altered

sense of time, subjects might experience merging of action and awareness, high concentration levels, sense of control over one's actions and the environment, unambiguous goals, loss of self-consciousness, clear and immediate feedback, and an autotelic (self-rewarding) feeling, experiencing the task as enjoyable and intrinsically rewarding [52,55].

Csikszentmihalyi and LeFevre [56] proposed that if subjects realize their work as an enjoyable experience, instead of seeing it as an obligation (i.e., as having lack of autonomy), they might have more frequent flow moments in the workplace. Flow is associated with positive mood [50], work engagement [57], and job satisfaction (which is a proxy of work-related happiness) [58], these contributing to high job performance of the employees [59], ultimately leading to increased productivity and economic revenue in for-profit organizations together with employees' personal growth. Considering that the use of self-leadership constructive thought patterns and natural reward strategies both promote the cognitive and emotional framing of work as an enjoyable experience, one would expect that evidence about the relationship between self-leadership and frequency of flow events in the workplace would be robust. As far as we are aware of, this theoretical inference has not been tested previously.

Self-leadership training has the potential to leverage psychological job-related resources so that individuals are autonomously motivated to respond to job demands, by transforming appraised energy draining job *hindrance stressors* into growth promoting job *challenge stressors* [60], even though many of the involved tasks might not be intrinsically enjoyable [16].

Mastering self-leadership as a competence includes using strategies that enable participants to: positively frame the task demands and suppress task negative issues (natural reward strategies); replace negative with positive beliefs concerning task demands (constructive thought pattern strategies); and establish self-rewards for goals defined according to own motivation and resources (behavior focused strategies). By strengthening individuals' set of self-leadership skills, a balanced equilibrium between perceived job challenges and personal skills may be anticipated and adequately tackled through self-development strategies, allowing the individual to better match both and, therefore, fulfilling a positive work condition, according to the JD-R model [8,16]. If job demands are more often matched by psychological job resources due to the development of self-leadership training, the number of flow [50] episodes at work should increase. This constitutes our second hypothesis. As job demands tend to fluctuate over time, depending on ad-hoc business needs, we expect that by applying self-leadership skills, individuals can adequately deploy their resources to maintain an optimal work experience, thus achieving and remaining in flow more often during the workday. If the self-leadership development increases the self-leadership skills measured over time, we expect our subjects to also increase their ability to match fluctuating job demands with their individual resources more effectively, therefore achieving more frequently flow experiences.

**Hypothesis 2** Self-leadership variation across time is associated with the variation of flow frequency in the workplace.

Flow experiences can be seen as episodic phenomena contributing to an overall eudaimonic state. Because flow is associated with positive mood [50], it is arguable that the occurrence of flow episodes might correlate with happiness when this construct is assessed. As the flow experience is an outcome of the adequate deployment of individual resources for the existing job demands at any given moment, individuals who are more frequently in flow will likely have more experiences of performing tasks successfully without boredom or distress, therefore reporting higher happiness in the workplace. If flow experiences become more frequent, we expect them to impact the immediate affective happiness state of the individuals [42] as measured by experience sampling methods. As such, we predict:

**Hypothesis 3** Flow is positively associated with happiness in the workplace, across time at the within-person level.

**Hypothesis 4** The effect of self-leadership on happiness is mediated by flow.

By considering the internal and external "conditions and processes that contribute to the flourishing or optimal functioning of people, groups, and institutions" [61] self-leadership theory is grounded in positive psychology. As such, it is counterintuitive that subjective well-being constructs, such as flow and happiness, have seldom been explored in the self-leadership literature [5,6]. This is a relevant area of study, going beyond the above-mentioned positive effects of self-leadership on psychological well-being (meaning lower levels of psychological suffering) and job performance objective indicators. More specifically, the associations between self-leadership and the experience of flow at work, and

between self-leadership and happiness at work, have not been established. Even if it is reasonable to establish a positive association between self-leadership and job satisfaction (as discussed earlier), this is not equal to assume that self-leadership has a positive impact on happiness: one can be satisfied with the work (job satisfaction) but not being happy with it, due to the role of positive affective and psychological functioning as more dispositional components of happiness, less dependent of contextual variables [62]. The study we present here aims to contribute to a better understanding of the association of self-leadership with flow and happiness in a longitudinal perspective, using a data collection process that enables fine-grain monitoring of flow and happiness: the experience sampling method. This allowed the observation of the co-evolution of self-leadership competencies and of flow and happiness momentary status.

## Materials and methods

### Research context

This study followed a longitudinal observational design, conducted in real-world organizational context. Participants were middle managers from different departments of a company in the fast-moving consumer goods (FMCG) sector. Participants who voluntarily enrolled in a self-leadership development program were offered the possibility of taking part of a self-monitoring component interlinked to the program. This procedure followed the common ethical good practices: before enrolling in the program effectiveness monitoring process, all participants were informed that their participation was voluntary and that it was not mandatory to take part in it to benefit from the training intervention, being free to inter-rupt their participation at any moment. Furthermore, information concerning the monitoring process and its objectives, data collection, storage and protection processes, and data confidentiality was provided both in-person and on the landing page for download of the smartphone app used for data collection (see *Procedure, measures of self-leadership, flow and happiness*). Participants were enrolled in the monitoring process after providing their written informed consent and before downloading the smartphone app for data collection. Data was collected between May 2$^{nd}$ 2016 (start of data collection for group 1) and February 7$^{nd}$ 2020 (end of data collection for group 7). Data confidentiality was ensured with pseudo-anonymization. No author had access to information that could identify individual participants. The data set was accessed on November 3$^{rd}$, 2022, to perform the analysis reported. This retrospective analysis of real-world archived data was approved by the Ethics and Deontology Committee of the Scientific Council of the Faculty of Psychology of the University of Lisbon.

### Self-leadership training course

A self-leadership blended-training program, i.e., in-room face-to-face and online training, was implemented following Marques-Quinteiro et al. [12]. The training reported in this paper was developed in 2013 and implemented in 2014 for that specific study, although its results were only published in 2018. Participants were randomly assigned to seven experimental groups that would engage in the self-leadership training (small-size training groups; all receiving the same training). Training of one group only started when the training and data collection of the previous one was completed. The training program was organized into four modules. Module 1 was about behavior-focused strategies; module 2 dealt with constructive thought pattern strategies; module 3 referred to natural rewards strategies; and module 4 was a consolidation of the previous modules. This last module was aimed at evaluating how successful the participants were at the implementation of self-leadership in the workplace, clarifying any doubts on the use of the three types of self-leadership techniques, and reinforcing implementation and impact through the application of already learned self-leadership skills to new situations or in different combinations than the ones originally trained [12,63]. Time interval between modules was of 2–3 weeks depending on the group. Each training module had a 4-hour in-room training session, followed by e-learning lessons, implementation exercises and virtual coaching through the learning platform, by an experienced trainer (for a detailed description of the training procedure, see Marques-Quinteiro et al. [12]).

## Participants

Sample characterization is provided in Table 1. Participants in the study were 64 middle-managers of a FMCG company. More than half of the respondents were male (70.3%). The mean age of participants was 41.05 (±7.55) years; concerning educational level, 45.3% of the respondents completed a bachelor's degree, whereas 7.8% had completed high school.

Concerning job-related variables, mean seniority in the company and in the current function was 10.17 (±8.09) years and 3.44 (±3.73) years, respectively. Mean weekly overtime was 6.34 (±5.47) hours. Participants reported different working contract types with different working schedule flexibility, from low to highly flexible working hours.

**Table 1. Participants' sociodemographic and work-related characterization.**

|  | Male (*n*=45) | Female (*n*=19) | Total (*N*=64) |
|---|---|---|---|
| **Age** (years) | | | |
| Range | 29−60 | 27−53 | 27−60 |
| Mean (SD) | 42.73 (7.28) | 37.05 (6.79) | 41.05 (7.55) |
| Median (IQR) | 43.00 (38.00−46.00) | 36.00 (32.50−40.00) | 42.00 (35.00−44.25) |
| **Age group (years)**, n (%) | | | |
| ≤ 40 years | 15 (33.3%) | 15 (78.9%) | 30 (46.9%) |
| > 40 years | 30 (66.7%) | 4 (21.1%) | 34 (53.1%) |
| **Educational level**, n (%) | | | |
| High school | 5 (11.1%) | 0 (0.0%) | 5 (7.8%) |
| Bachelor's degree | 21 (46.7%) | 8 (42.1%) | 29 (45.3%) |
| Postgraduate studies | 15 (33.3%) | 3 (15.8%) | 18 (28.1%) |
| Master's degree | 4 (8.9%) | 8 (42.1%) | 12 (18.8%) |
| **Seniority in the company**, years | | | |
| Range | 0−31 | 2−30 | 0−31 |
| Mean (SD) | 10.74 (8.41) | 8.82 (7.31) | 10.17 (8.09) |
| Median (IQR) | 9.00 (4.00−16.00) | 6.00 (5.00−9.00) | 8.00 (4.88−12.50) |
| **Seniority in the function**, years | | | |
| Range | 0−15 | 0−5 | 0−15 |
| Mean (SD) | 4.26 (4.13) | 1.51 (1.19) | 3.44 (3.73) |
| Median (IQR) | 2.00 (1.00−6.00) | 1.00 (0.88−2.00) | 2.00 (1.00−5.00) |
| **Weekly overtime**, hours | | | |
| Range | 0−22 | 0−15 | 0−22 |
| Mean (SD) | 6.53 (5.71) | 5.89 (5.00) | 6.34 (5.47) |
| Median (IQR) | 5.00 (0.00−10.00) | 5.00 (0.00−10.00) | 5.00 (0.00−10.00) |
| **Weekly overtime**, n (%) | | | |
| ≤ 5 hours | 26 (57.8%) | 11 (57.9%) | 37 (57.8%) |
| > 5 hours | 19 (42.2%) | 8 (42.1%) | 27 (42.2%) |
| **Working schedule flexibility**, n (%) | | | |
| None | 0 (0.0%) | 0 (0.0%) | 0 (0.0%) |
| Low | 1 (2.2%) | 0 (0.0%) | 1 (1.6%) |
| Some | 18 (40.0%) | 9 (47.4%) | 27 (42.2%) |
| High | 20 (44.4%) | 8 (42.1%) | 28 (43.8%) |
| Total | 6 (13.3%) | 2 (10.5%) | 8 (12.5%) |

SD, Standard deviation; IQR, Interquartile range

## Data collection procedure

Two complementary approaches were followed for data collection: (1) email surveys, launched in four different moments (longitudinal approach) within one week after training modules one to three, and two to four months after the last (fourth) module finished and (2) experiencing sampling method (ESM) throughout the intervention program. Both email surveys and ESM questionnaires were completed during working hours (9 a.m. to 8 p.m.) on business days. Data on self-leadership and dispositional flow were collected via email surveys; situational flow and happiness were collected through ESM.

Experiencing sampling method is a diary technique that consists of asking participants to complete a short self-report questionnaire about their experience at a given moment in time, several times within a defined period [64–68]. When psychological phenomena are expected to be highly dynamic, with strong variation in short periods, ESM is a powerful assessment because it is a quick but randomly sampled self-report throughout the days – repeated assessments in subjects' daily lives [64–68]. This data collection approach has been used to study happiness [38,69], flow (e.g., Ceja and Navarro [70]; Csikszentmihalyi and LeFevre [56]), among other psychological phenomena (e.g., Sitko, Varese, Sellwood, Hammond, and Bentall [71]; Thewissen et al. [72]). Flow experience is a non-ergodic process [70,73], varying between individuals and over time. Indeed, Ceja and Navarro [70] found that nonlinear behavioral patterns are associated with the flow state. This calls for flow research in the workplace to go beyond between-subjects differences, also assessing within-subject differences, to capture flow intra-subject variability over time. Again, ESM was used for this purpose.

Experiencing sampling method data collection was made via an online smartphone app, thrice a day, between two to three months. Participants downloaded the app (iSURVEY° for iOS, droidSURVEY° for android platforms) to their smartphones and authenticated the device within a common centralized survey account using a password. Remote assistance by the research team for this process was provided when required. After downloading the app and authenticating the device, the survey questions were then automatically downloaded to participants' devices. The app allowed participants to register their responses even in the absence of an internet connection. All replies were saved on each device until a new internet connection was established, at which point all responses were automatically uploaded to the common database. Individual answers to each item were automatically recorded with the person's identification code (as registered by them in the system) together with a time stamp.

When participants first opened the app, the survey appeared on the screen. They were then asked to answer each question with a 5-level slider (five-point Likert-type scale), which would display either graphically and/or numerically, depending on the question. Item presentation always followed the same order: flow first, happiness second. The average answering time was 40 seconds. Participants were asked to complete the ESM queries three times throughout three different periods of the working day: the first from 9 a.m. to 1 p.m., the second from 1 p.m. to 5 p.m., and the third from 5 p.m. to 8 p.m. A message automation tool was used to send individual reminders three times a day, once in each period, to remind participants to respond. The reminders were set to go off randomly inside each of these specific periods. Participants could access the app and respond to the survey straight away or within the remainder time of that period. Participants would be excused from answering if they were on vacation, weekend or otherwise unavailable. Consequently, the results would only reflect data collected during working hours on business days (ensuring a homogenous environment of data collection).

Responses via the app were reviewed by the research team every week and, when necessary, more reminders were issued (via email) to emphasize the importance of getting daily responses.

All collected data are available, together with the R syntax needed to reproduce the results, on Zenodo platform (https://doi.org/10.5281/zenodo.15395468).

## Measures

***Self-leadership.*** A short version of the Abbreviated Revised Self-Leadership Questionnaire [74] adapted and validated for the Portuguese population by Marques-Quinteiro et al. [75] was used to measure self-leadership. Each of the 13 items is assessed on a five-point Likert-type scale, ranging from 1 (*never*) to 5 (*always*). The final score is obtained by averaging the scores of each item (high values expressing more frequent use of self-leadership strategies).

***Flow.*** The formulation of the questions related to flow assessment differed between the e-mail surveys and the ESM approach. For the e-mail survey, the questions assessed work-related flow in a general (dispositional) format: (1) How challenging do I find the activities I am performing as part of my role? (1 = *a little;* 5 = *a lot*), (2) What is my skill level for performing these activities? (1 = *a little;* 5 = *a lot*), (3) How much do I enjoy doing these activities? (1 = *a little;* 5 = *a lot*), (4) How interesting are these activities? (1 = *not interesting at all;* 5 = *very interesting*), and finally, (5) How quickly does time pass while I am doing these activities? (1 = *time passes very slowly;* 5 = *time passes very fast*) [54,73]. In the ESM approach, the formulation of the items assessed flow in a rather situational (i.e., momentum-specific oriented) perspective: (1) The task I am doing at this moment is challenging; (2) I have the skills to perform this activity; (3) I enjoy doing this activity; (4) This activity is interesting; and finally (5) Time passes quickly while I am doing this activity. The answer to each of these items was done on a 5-point Likert scale (1 = *I don't agree;* 5 = *I totally agree*). The final flow score is obtained by averaging the scores of each item (higher values expressing more intense flow experiences).

***Happiness.*** Situational happiness (ESM) was assessed using two items from the Subjective Happiness Scale adapted and validated for the Portuguese population [76]: participants were asked to rate their happiness in the present time (*At this moment, I feel…*) on a five-point Likert-type scale, ranging from 1 (*not a very happy person*) to 5 (*a very happy person*), and in comparison with their colleagues (*Compared to most of my peers, I consider myself…*), also in a five-point Likert-type scale, ranging from 1 (*less happy than them*) to 5 (*happier than them*). The final score is obtained by averaging the scores of each item (higher values expressing more happiness self-reporting).

***Sociodemographic variables.*** Participants were asked to register a small set of sociodemographic (sex, age, and educational level) and job-related data: seniority in the workplace, average working hours per week (and weekly overtime hours), working contract type, and working schedule flexibility).

## Statistical analysis

All analyses were conducted using R version 3.6.3 [77]. Descriptive statistics on sociodemographic and work-related variables are given as mean and standard deviation, median, and interquartile range for quantitative variables; absolute and relative (%) frequencies are provided for qualitative variables.

Linear mixed models [78] were used to assess changes over time in self-leadership, flow and happiness, and to test hypotheses 1–3. Linear mixed models were estimated by maximum likelihood using all data available. These models have been widely used to analyse data collected through ESM (e.g., Sitko et al. [71]; Thewissen et al. [72]; Vork et al. [79]), being particularly suitable for analysing this type of data because they account for the hierarchical structure of the data (observations nested within individuals), as well as for differences between individuals in the number and spacing across observations [65]. Two-level models with a random intercept by individual were used for analysing data from email surveys. Differences between the four moments of assessment in self-leadership, situational happiness and situational and previous 15-days flow assessment were evaluated by t-tests with *p*-value adjustment (Tukey method) for multiple comparisons. Before being introduced in these two-level models, happiness values obtained through ESM were averaged across days for each individual, until the delivery date of the email survey (time points when self-leadership indicators were assessed). For example, if the first and second e-mail assessments occurred at day 10 and 20, respectively, all observations up to day 10, and those from day 11 to day 20 were averaged; these averages correspond to time of assessment 1 and time of assessment 2, respectively. The same procedure was followed for the last two assessments; this allowed the calculation of four happiness values, each value corresponding to one out of the four assessments via email survey. To model and compare situational and dispositional flow assessment, the procedure described for happiness was also employed to the values of situational flow obtained through ESM two-level models, happiness values obtained through ESM were averaged across days for each individual, until the delivery date of the email survey (time points when self-leadership indicators were assessed). For example, if the first and second e-mail assessments occurred at day 10 and 20, respectively, all observations up to day 10, and those from day 11 to day 20 were averaged; these averages

correspond to time of assessment 1 and time of assessment 2, respectively. The same procedure was followed for the last two assessments; this allowed the calculation of four happiness values, each value corresponding to one out of the four assessments via email survey. To model and compare situational and dispositional flow assessment, the procedure described for happiness was also employed to the values of situational flow obtained through ESM.

To test the effect of between- and within-subject self-leadership differences in happiness (hypothesis 1) and in situational and dispositional flow (hypothesis 2), person-mean ($X_{cb}$) and person-mean centered ($X_{cw}$) variables were calculated. Briefly, the global mean, $\overline{X}$, was first subtracted from $X$ to obtain the centered grand mean, $X_{cg}$. By averaging $X_{cg}$ across observations for each individual, $X_{cb}$ was obtained. This varies between individuals, but not within-individual, and denotes how far the average value per individual is from the grand mean. Finally, $X_{cw} = X_{cg} - X_{cb}$ measures the deviation of each observation from the typical value of each individual. For regression models, $X_{cb}$ predicts differences between individuals in $Y$ given between-subject differences in $X$, whereas $X_{cw}$ predicts intra-individual changes in $Y$ based on within-subject changes in $X$ [80]. Sociodemographic and work-related variables were dichotomized and included as fixed effects in the models, along with time of evaluation (categorical variable: times 1–4).

The association between situational flow and situational happiness (hypothesis 3) was tested using the complete ESM dataset (not averaging the data). In this case, a three-level model (observations nested within days nested within individuals) was used, assuming a random intercept for each level of 'individual' and for each level of 'day' nested within 'individual'. 'Day' was defined as the number of working days counted since the first answer provided via the smartphone app by each individual and was included in the model to control for the effect of time (quantitative variable); 'time slot' corresponded to the time slot during working hours when the subject responded to ESM questionnaires (categorical variable); between- and within-subject flow, along with sociodemographic and work-related variables, were included in the model as fixed effects. Finally, four-level models (observations > days > assessments > individuals) were used to estimate situational flow and happiness as a function of self-leadership (Supporting S1 Table in supporting information). Random slopes were tested for all models. Unless otherwise indicated, the principle of parsimony was applied, and non-significant slopes were removed from the model.

Effect sizes are provided as regression coefficients ($b$) and 95% confidence intervals (CI). The Satterthwaite's degrees of freedom method [81] was used to calculate $p$ values; statistical significance was set to $\alpha = .05$. For each model, the conditional $R^2$ refers to the variance explained by both random and fixed effects, whereas the marginal $R^2$ refers to the variance explained by fixed effects only.

Within-subject mediation analyses [82,83] were conducted to test the mediation effect of flow (hypothesis 4) on the relationship between self-leadership (independent variable, X) and happiness (dependent variable, Y). These variables were all measured at level 1 (i.e., time of assessment level); so lower level (1-1-1) mediation models were specified, including a random intercept by individual and random slopes for the paths $a$ (X→M), $b$ (M→Y) and $c'$ (X→Y). The random effects were allowed to correlate, while in the fixed part of the model the variables X, M and Y were person-mean centered and the time of assessment was included as predictor of M and Y. Sociodemographic and work-related variables were also tested as covariates, being kept as final models the most parsimonious ones. The total effect of self-leadership on happiness is given by $c = c' + ab + \sigma_{a_j b_j}$, where $c'$ is the direct effect of X on Y; $a$ and $b$ are the average X→M and M→Y effects, respectively; and $\sigma_{a_j b_j}$ is the covariance of between-subjects differences in those effects. The mediated effect, known as the indirect effect of X on Y through M, is given by $me = c - c'$, for which 95% CI were obtained following a Monte Carlo approach [84]. This effect was considered significant if the interval did not contain the value zero.

Psychometric properties of the self-leadership and flow instruments were assessed by calculating multilevel reliability, between-person reliability ($R_{KF}$) and within-person reliability ($R_C$) [85], using data from the email surveys.

## Results

The self-leadership instrument revealed good psychometric properties in this study, when estimated as between-person reliability ($R_{KF} = 0.97$), and fair properties when estimated within-person reliability ($R_C = 0.51$). The instrument used

to measure flow also showed good psychometric properties, when estimated as between-person reliability ($R_{KF}$ = 0.91), and fair when estimated within-person reliability ($R_C$ = 0.63). Fig 1 and Supporting S2 Table in supporting information provide an overview of the evolution of happiness, flow and self-leadership throughout the study period. Most participants responded to all assessments (81.3%), 17.2% failed to answer once, and one individual completed only one out of four assessments. Hence, 242 out of 256 possible e-mail survey observations were available for analysis (corresponding to a reasonable percentage of missing data – 5.5% - not requiring data imputation [86]). Regarding the ESM, a total of 4383 observations were collected by the app and included in the analysis.

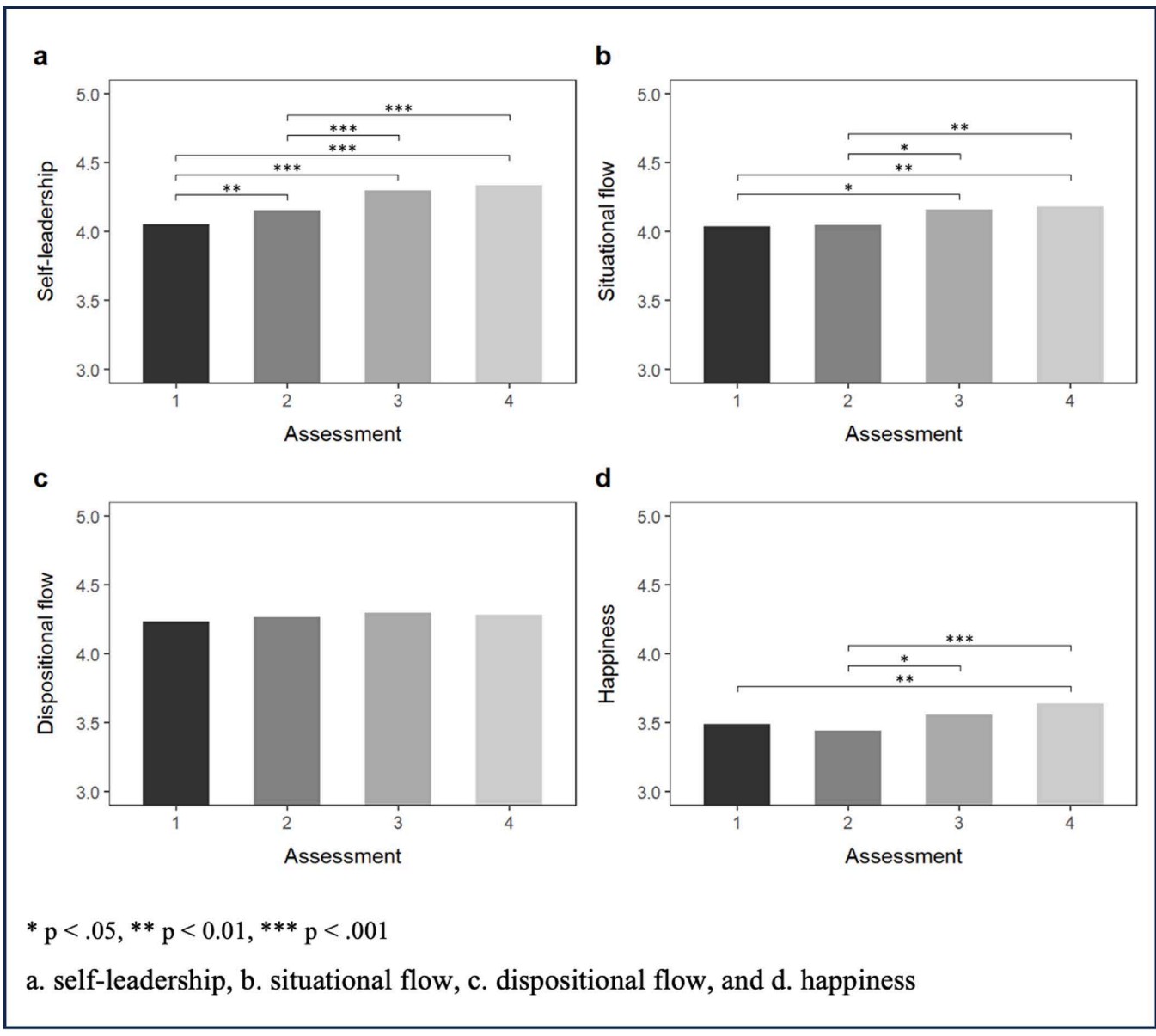

**Fig 1. Estimated means for each time of assessment (1 to 4).**

Mean self-leadership significantly increased over time since the first assessment (4.05, 95% CI: 3.95−4.16) to the last one (4.34, 95% CI: 4.24−4.44; $p < .001$; Fig 1a). A similar pattern was obtained for situational flow assessed through ESM (mean situational flow $_{time\ 1}$=4.04, 95% CI: 3.92−4.15; mean situational flow $_{time\ 4}$=4.18, 95% CI: 4.07−4.30; $p=.003$; Fig 1b), but not for dispositional flow assessed through e-mail survey (mean dispositional flow $_{time\ 1}$=4.24, 95% CI: 4.12−4.35; mean dispositional flow $_{time\ 4}$=4.28, 95% CI: 4.17−4.40; $p=.856$; Fig 1c). Mean happiness also increased over the study period (mean happiness $_{time\ 1}$=3.49, 95% CI: 3.38−3.60; mean happiness $_{time\ 4}$=3.64, 95% CI: 3.52−3.75; $p=.004$; Fig 1d). No differences between measurements taken at time 3 and time 4 were obtained for self-leadership, flow, or happiness (Table 2).

## Self-leadership and flow

Regarding hypothesis 2, increased within- ($b=0.41$, 95% CI: 0.14−0.68) and between-subjects ($b=0.63$, 95% CI: 0.44−0.82) self-leadership was associated with higher dispositional flow assessed through the e-mail survey. The only

**Table 2. Differences between times of assessment for self-leadership, dispositional and situational flow, and happiness.**

| Contrast between time | Estimate | SE | Lower CL | Upper CL | p |
|---|---|---|---|---|---|
| **Self-leadership** | | | | | |
| 1–2 | −0.10 | 0.03 | −0.18 | −0.02 | **.007** |
| 1–3 | −0.24 | 0.03 | −0.32 | −0.16 | **<.001** |
| 1–4 | −0.28 | 0.03 | −0.36 | −0.20 | **<.001** |
| 2–3 | −0.14 | 0.03 | −0.22 | −0.06 | **<.001** |
| 2–4 | −0.18 | 0.03 | −0.26 | −0.10 | **<.001** |
| 3–4 | −0.04 | 0.03 | −0.12 | 0.04 | .589 |
| **Situational flow** | | | | | |
| 1–2 | −0.01 | 0.04 | −0.12 | 0.09 | .989 |
| 1–3 | −0.12 | 0.04 | −0.23 | −0.02 | **.017** |
| 1–4 | −0.14 | 0.04 | −0.25 | −0.04 | **.003** |
| 2–3 | −0.11 | 0.04 | −0.21 | −0.00 | **.039** |
| 2–4 | −0.13 | 0.04 | −0.23 | −0.03 | **.008** |
| 3–4 | −0.02 | 0.04 | −0.13 | 0.08 | .954 |
| **Dispositional flow** | | | | | |
| 1–2 | −0.03 | 0.06 | −0.18 | 0.12 | .952 |
| 1–3 | −0.06 | 0.06 | −0.21 | 0.09 | .703 |
| 1–4 | −0.05 | 0.06 | −0.20 | 0.10 | .856 |
| 2–3 | −0.03 | 0.06 | −0.18 | 0.12 | .949 |
| 2–4 | −0.02 | 0.06 | −0.17 | 0.14 | .994 |
| 3–4 | 0.02 | 0.06 | −0.14 | 0.17 | .992 |
| **Happiness** | | | | | |
| 1–2 | 0.05 | 0.04 | −0.07 | 0.16 | .713 |
| 1–3 | −0.07 | 0.04 | −0.18 | 0.04 | .343 |
| 1–4 | −0.15 | 0.04 | −0.26 | −0.04 | **.004** |
| 2–3 | −0.12 | 0.04 | −0.23 | −0.01 | **.035** |
| 2–4 | −0.19 | 0.04 | −0.31 | −0.08 | **<.001** |
| 3–4 | −0.08 | 0.04 | −0.19 | 0.04 | .301 |

SE, Standard Error; CL, Confidence Limit

Statistically significant values indicated in bold

sociodemographic variable that was associated with dispositional flow was educational level, with postgraduate studies being significantly associated with lower levels of dispositional flow ($b = −0.18$, 95% CI: −0.32−−0.04; Table 3; Fig 2).

Concerning situational flow, only between-subjects self-leadership had a significant effect ($b = 0.42$, 95% CI: 0.16−0.68). No association between sociodemographic and work-related variables and situational flow was detected (Table 3; Fig 2). A similar effect of between-subjects self-leadership on situational flow assessed using data available from ESM was obtained ($b = 0.43$, 95% CI: 0.18−0.69; Supporting S1 Table in supporting information).

## Self-leadership and flow are associated with happiness

Regarding hypothesis 1 (Table 3; Fig 2), higher between-subjects self-leadership ($b = 0.44$, 95% CI: 0.19−0.69) was associated to higher happiness. Educational level and time of assessment were also associated with happiness measured through email survey; holding a postgraduate course was significantly associated with lower levels of happiness ($b = −0.19$, 95% CI: −0.37−−0.01) and happiness was significantly higher in the last moment of assessment ($b = 0.12$, 95% CI: 0.02−0.23) compared with the first one. An effect of comparable magnitude of between-subject self-leadership on happiness was also observed in four-level models ($b = 0.46$, 95% CI: 0.21−0.70; Supporting S1 Table in supporting information).

The association between situational flow and happiness was also investigated (hypothesis 3), and both within- ($b = 0.61$, 95% CI: 0.55−0.67) and between-subjects ($b = 0.53$, 95% CI: 0.32−0.74) flow were positively and significantly associated to happiness. The time slot 5–8 p.m. was associated with higher levels of happiness than the morning time slot; no association between sociodemographic and other work-related variables and happiness measured through ESM was observed (Table 4; Fig 3). The individual variation of situational flow and happiness is provided as Supporting S1 Fig in supporting information.

Regarding hypothesis 4, data do not support a within-subject indirect effect of self-leadership on happiness through dispositional flow ($me = 0.02$, 95% CI: −0.03−0.09). Higher self-leadership is associated to higher dispositional flow

**Table 3. Parameter estimates and 95% confidence intervals (CI) for the effect of between- and within-subject self-leadership on dispositional and situational flow, and happiness, controlled for timeslot of data collection, sociodemographic and work-related variables.**

| Fixed effects | Dispositional flow | | | Situational flow | | | Happiness | | |
|---|---|---|---|---|---|---|---|---|---|
| | Estimates | 95% CI | *p* | Estimates | 95% CI | *p* | Estimates | 95% CI | *p* |
| (Intercept) | 4.32 | 4.11–4.52 | **<.001** | 3.89 | 3.62–4.15 | **<.001** | 3.60 | 3.34–3.85 | **<.001** |
| Assessment: 2 | −0.01 | −0.12–0.11 | .888 | −0.02 | −0.10–0.06 | .634 | −0.06 | −0.15–0.03 | .175 |
| Assessment: 3 | −0.04 | −0.17–0.09 | .557 | 0.06 | −0.03–0.16 | .169 | 0.04 | −0.06–0.14 | .467 |
| Assessment: 4 | −0.07 | −0.20–0.07 | .312 | 0.09 | −0.01–0.18 | .083 | 0.12 | 0.02–0.23 | **.022** |
| Self-leadership between-subjects | 0.63 | 0.44–0.82 | **<.001** | 0.42 | 0.16–0.68 | **.002** | 0.44 | 0.19–0.69 | **.001** |
| Self-leadership within-subjects | 0.41 | 0.14–0.68 | **.003** | 0.17 | −0.02–0.36 | .085 | 0.08 | −0.13–0.28 | .479 |
| Sex: female | 0.16 | −0.00–0.33 | .053 | 0.17 | −0.05–0.40 | .132 | −0.03 | −0.24–0.19 | .795 |
| Age group: >40y | 0.04 | −0.11–0.20 | .572 | 0.13 | −0.08–0.34 | .217 | −0.12 | −0.32–0.08 | .241 |
| Educational level: postgraduate studies | −0.18 | −0.32 − −0.04 | **.012** | −0.06 | −0.25–0.13 | .534 | −0.19 | −0.37 − −0.01 | **.042** |
| Seniority in the company: ≥ 10y | 0.04 | −0.11–0.19 | .596 | 0.03 | −0.18–0.24 | .771 | 0.09 | −0.10–0.29 | .340 |
| Seniority in the function: >2y | 0.05 | −0.10–0.21 | .492 | 0.16 | −0.05–0.37 | .133 | 0.05 | −0.15–0.25 | .633 |
| Weekly overtime: >5h | 0.01 | −0.12–0.15 | .829 | 0.13 | −0.06–0.32 | .180 | 0.07 | −0.11–0.25 | .415 |
| Working schedule flexibility: high[a] | −0.08 | −0.23–0.06 | .239 | −0.07 | −0.27–0.12 | .458 | −0.04 | −0.22–0.15 | .681 |
| Number of observations | 242 | | | 236 | | | 236 | | |
| Marginal $R^2$ / Conditional $R^2$ | 0.30/ 0.52 | | | 0.22/ 0.77 | | | 0.22/ 0.73 | | |

[a] High flexibility includes high and total working schedule flexibility

Statistically significant values indicated in bold

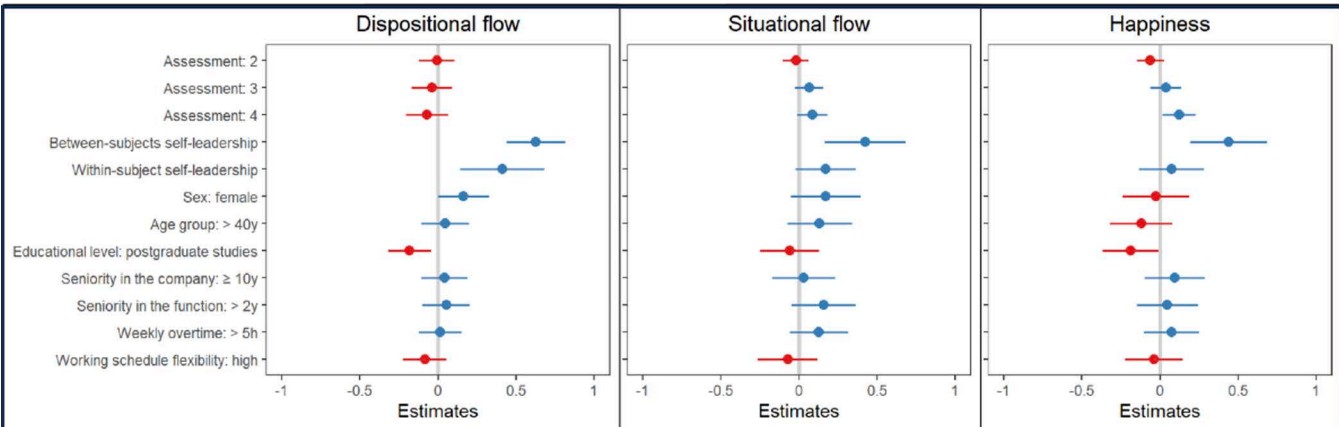

Dots and lines are estimates for each predictor and the corresponding 95% confidence intervals (IC). 95% CI that does not cross the vertical axis indicates statistically significant variables at 5%. Blue and red colors are independent variables with positive and negative effects in the dependent variable, respectively. High flexibility includes high and total working schedule flexibility. * p < .05, ** p < 0.01, *** p < .001

**Fig 2. Results of the linear mixed models for situational and dispositional flow, and happiness.**

**Table 4. Parameter estimates and 95% confidence intervals (CI) for the effect of between- and within-subject situational flow on happiness, while controlling for time slot of data collection, sociodemographic and work-related variables.**

| Fixed effects | Happiness | | |
|---|---|---|---|
| | Estimates | 95% CI | p |
| (Intercept) | 3.68 | 3.44–3.91 | **<.001** |
| Day | −0.00 | −0.00–0.00 | .979 |
| Time slot: 1–5 p.m. | 0.01 | −0.02–0.04 | .595 |
| Time slot: 5–8 p.m. | 0.03 | 0.00–0.06 | **.043** |
| Between-subjects flow | 0.53 | 0.32–0.74 | **<.001** |
| Within-subject flow | 0.61 | 0.55–0.67 | **<.001** |
| Sex: female | −0.09 | −0.30–0.11 | .352 |
| Age group: > 40y | −0.16 | −0.35–0.03 | .091 |
| Educational level: postgraduate studies | −0.14 | −0.31–0.03 | .097 |
| Seniority in the company: ≥ 10y | 0.02 | −0.16–0.20 | .826 |
| Seniority in the function: > 2y | −0.02 | −0.20–0.17 | .863 |
| Weekly overtime: > 5h | 0.06 | −0.10–0.23 | .459 |
| Working schedule flexibility: high | 0.01 | −0.16–0.18 | .897 |
| Number of observations | 4383 | | |
| Marginal $R^2$ / Conditional $R^2$ | 0.40/ 0.73 | | |

[a] High flexibility includes high and total working schedule flexibility

Statistically significant values are in bold

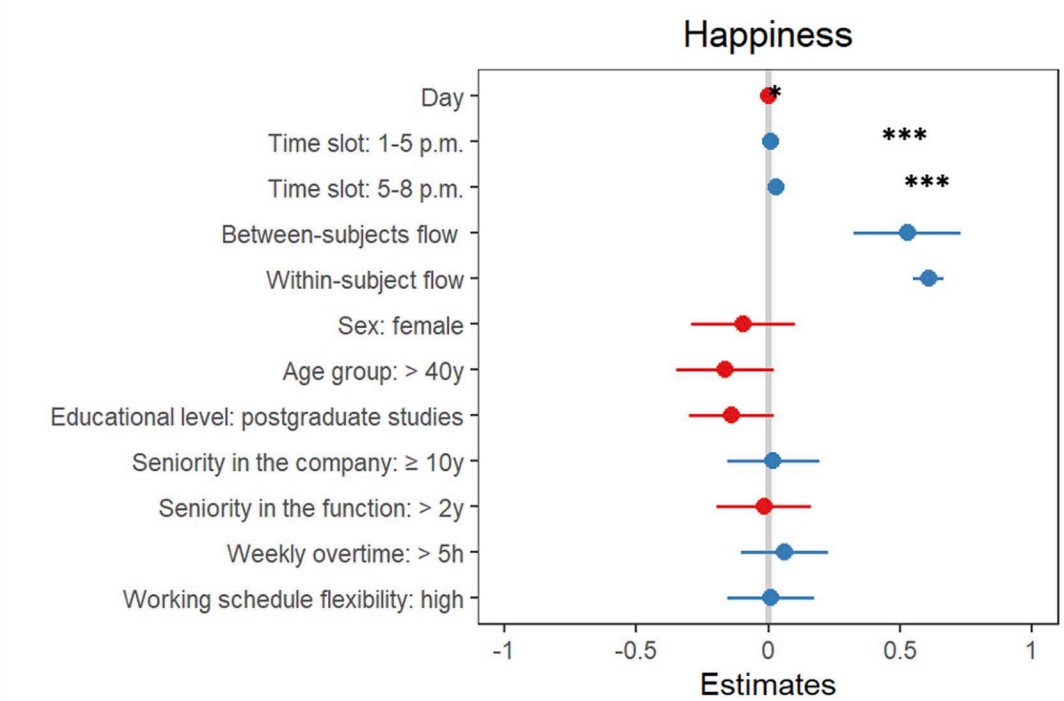

**Fig 3. Results of the linear mixed models for happiness.**

within subjects ($a = 0.37$, $p = 0.013$); however, neither dispositional flow ($b = 0.05$, $p = 0.479$) nor self-leadership ($c' = 0.03$, $p = 0.751$) are statistically related to happiness when accounting for the effect of each on the other (Fig 4, Model 1). The mediation effect of situational flow on the relationship between self-leadership and happiness was also non-significant ($me = 0.10$, 95% CI: $-0.04 - 0.25$). In this model, while self-leadership is not associated with situational flow ($a = 0.19$, $p = 0.093$) or happiness ($c' = -0.06$, $p = 0.572$), higher situational flow was related to increased happiness ($b = 0.64$, $p < 0.001$), for individuals with the same level of self-leadership (Fig 4, Model 2). The effect of time of assessment in flow and happiness was controlled in both models. Sociodemographic and work-related variables were not significant and, therefore, were removed from the models.

## Discussion

This study addressed the possibility that self-leadership variation across time is positively associated with employees' positive affect in the workplace. Findings herein suggest that self-leadership variation across time is positively and significantly associated with experiencing flow and happiness in the workplace. To our knowledge, this was the first attempt to comprehensively understand the relationship between development of self-leadership through a longitudinal approach,

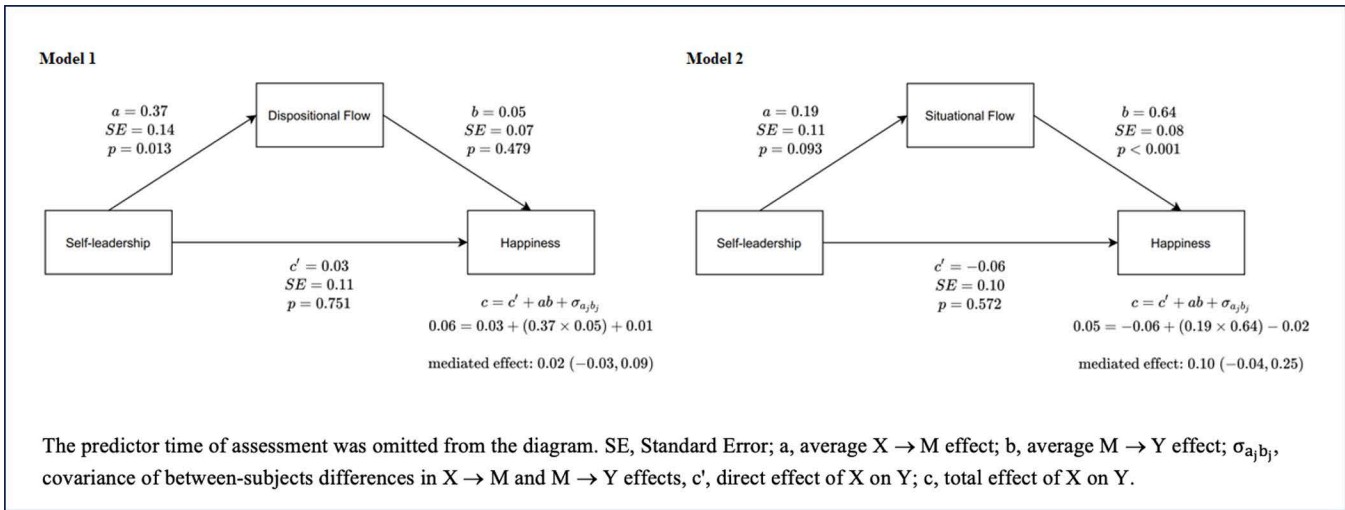

**Fig 4. Within-subject mediation: mediated effect (95% CI) of self-leadership (X) on happiness (Y) through dispositional flow (Model 1) or situational flow (Model 2) as mediator (M).**

using a self-leadership development program, and (1) situational and dispositional experiences of flow at workplace and (2) happiness in the workplace, both over time.

## Self-leadership and positive affect over time

As expected, subjects' self-leadership competencies significantly increased over time within the context of a self-leadership development program, which embodies growing evidence in favor of self-leadership development programs in the work context (e.g., Furtner et al. [30]; Marques-Quinteiro et al. [12]; Neck and Manz [13]; Stewart et al. [87]; Unsworth and Mason [29]). Interestingly, self-leadership competencies did not significantly change between assessment moments 3 and 4 of our study. This makes sense because no new self-leadership competencies were learned at module 4 (which anticipated the fourth assessment-moment). Indeed, participants had already received training in the three domains of self-leadership, i.e., behavior-focused strategies (in module 1), constructive thought pattern strategies (in module 2) and natural reward strategies (in module 3). Assessment moment was sent two to four months after the consolidation module, that did not provide new self-leadership techniques. This observed stability of self-leadership competencies is in line with data collected in other studies that indicate that these self-leadership skills remained stable over time [12,13].

Regarding positive affect, both happiness and situational flow did not significantly change between assessment moments 3 and 4, whereas dispositional flow remained stable over the study period. Two arguments should be considered here. First, the characteristics of the work environment should be considered. Data were collected during working hours (9 a.m. to 8 p.m.) on business days and, thus, one might consider data collection environment to be quite homogenous. This was because the tasks being performed by each individual would be roughly the same with no shifts between working and leisure times. If data collection would have been performed during both working and leisure time, greater fluctuations in flow and happiness would be expected based on the non-ergodic nature of the former [70,73]. If the homogenous data collection environment might explain the results for dispositional flow, an additional element should be considered for happiness and situational flow, and this is the hypothesized impact of increased levels of self-leadership and its stability between assessment moments 3 and 4 of the self-leadership training. The absence of differences of flow and happiness between moments 3 and 4 is in line with the association between self-leadership and happiness (hypothesis 1),

as well as between self-leadership and flow (hypothesis 2). According to our hypotheses, not observing changes in self-leadership would result in no change in both happiness (hypothesis 1) and flow (hypothesis 2).

Also as expected, the acquisition of self-leadership strategies (i.e., behavior-focused strategies, constructive thought pattern strategies and natural reward strategies) was positively associated to experiences of flow, thus supporting hypothesis 2. This finding suggests that self-leadership training contributes to an improved balance between perceived skills and challenges in the workplace. Although the Job Demands and Resources model does not propose self-leadership as a tool for promoting employees' self-efficacy and, ultimately, employees' flow or happiness, it is difficult to figure out how, in the model, resources can adequately match demands without individuals' strategic self-regulation of such resources, considering personal goals (i.e., without effective self-leadership). As theoretically developed in the introduction section, self-leadership role in the job demands-resources perceived balance (by employees) can be a main active-principle of self-leadership associations with both the feeling of flow and happiness. Indeed, as conceptualized by Lovelace et al. [57], flow seems to be an outcome following the practice of self-leadership. This is a novel topic in the self-leadership literature [5] and, to our knowledge, this is the first contribution to assess this relationship in the real-world workplace.

As argued by Ceja and Navarro [70], a comprehensive analysis of well-being in the workplace measured as flow is not complete unless both between- and intra-individuals' variability are considered. Results herein add support to this claim. Between-subject self-leadership was associated to both situational and dispositional flow, whereas within-subject self-leadership was only associated to dispositional flow. These findings might be explained by the experimental approach here taken: data collection was always conducted during the working time on business days. By doing so, environmental variability associated with working versus leisure times was minimized and, differently from previous findings [54,70], a more stable balance between job resources and demands was detected.

Results herein support hypothesis 1: between-subject self-leadership variation across time was positively associated to happiness. Results herein add therefore to the growing body of literature on the positive association between self-leadership and happiness-related constructs (e.g., Neck and Manz [13]; Roberts and Foti [20]). Also, our data revealed that between- and within-subject flow variation was positively associated with happiness (hypothesis 3). In other words, a better (more frequent) fit between job demands and resources (associated, as described, with self-leadership strategies development), inherent to flow experiences, came out in our study as associated to happiness. There is scarce evidence on this topic from previous research; nevertheless, previous data from sports practitioners has also shown a positive influence of flow over happiness [88]. This is a novel finding with practical implications for for-profit organizations, to be discussed below. The association between self-leadership and flow deserves a more detailed reflection and empirical investigation. It is rather consensual that the main aim of self-leadership is to promote a balancing of job demands and resources according to the subjects' purpose and goals [12]. By using constructive thought patterns, behavior focused, and natural reward strategies, the subjects may reframe (cognitive relabelling) job hindrance stressors (for example: difficult tasks, work shifts, negative work environment, non-motivating activities, low pay, etc.) into job challenge growth-promoting factors (for example: experience opportunity, new skills to be acquired, filling industry gaps in the CV, exposure to key decision makers, etc.). These same self-leadership strategies can be deliberately used by the subjects to increase their job resources (for example: negotiating a flexible work schedule, planning big tasks in smaller bites according to a chosen work schedule, developing positive beliefs about professional growth, setting rewards according to personal goals, understanding own competency gaps and planning for their fulfilment, etc.). This can promote both a more objective (i.e., less emotionally charged) assessment of the job demands and resources and the individual's ability to balance them. As the flow experience is more likely when individuals are performing tasks at the top level of their current competence level but not above it, this requires a constant balance between the job demands and resources at any given moment, which can be provided by the deliberate use of self-leadership at work. In this sense, an increase in self-leadership will be associated with an increase in flow.

 

Hypothesis 4 was not confirmed by data: flow was not found to act as a mediator between self-leadership and happiness. It is important to recall that happiness was measured through ESM whereas flow experiences (tested as mediator) were measured through e-mail surveys after each training module (dispositional flow). This was done because situational flow (collected through ESM) was not found to be associated with self-leadership – which is a condition for mediation analysis. So, the data collected procedures of flow and happiness were not equal and this can eventually explain the absence of mediation.

The fact that flow experiences do not mediate the association between self-leadership and happiness can also be explained by the chosen measure of happiness, which follows a hedonic approach (based on Lyubomirsky and Lepper's tool) rather than a eudaimonic approach. Although we have found a positive association between flow and happiness, it is important to keep in mind that flow experiences do not necessarily imply hedonic happiness: as a matter of fact, flow experiences are theoretically more related with a eudaimonic perspective of happiness, in which meaningfulness of human acts play a central role [55]. An avenue to pursue in future studies is to study this potential mediation effect of flow by using a tool measuring eudaimonic well-being (e.g., Waterman et al. [89]).

### Self-leadership in the organizational context: practical implications for employees' positive affect

We believe that the outcomes of this study are highly relevant for organizations. Workers in the manufacturing industry, tourism and leisure activity, transportation, and real estate and housing are often affected by downsizing, (temporary) layoffs and job loss [90–92]. Changes to work arrangements, including working from home and working in shifts are more and more replacing office working, with numerous challenges for employees' subjective well-being. As such, training programs for developing and leveraging employees' self-leadership skills can be instrumental for achieving increased flow and happiness at work in organisations during challenging times.

This research also adds to previous studies by providing evidence that the three components of self-leadership are trainable and potentially sustained over time (e.g., Furtner et al. [30]; Lucke and Furtner [93]; Unsworth and Mason [29]) in the for-profit organisational context [12] and in line with the assumptions of the job demands and resources model. Moreover, it suggests the suitability of self-leadership training in the FMCG sector for achieving a perceived balance between job resources and demands and positive affect. In a recent study, Marques-Quinteiro et al. [12] concluded that self-leadership training was adequate for developing employees' adaptive performance and job satisfaction in the private banking sector during a crisis.

### Strengths and limitations

A comprehensive analysis of the results provided in this study benefits from the acknowledgment of its strengths and limitations. Concerning its strengths, to our knowledge, this study is the first of its kind to address the association between self-leadership and positive affect measured as flow and happiness in the for-profit real-world organizational setting (see Goldsby et al. [5]). Second, both between-subject and within-subject variability in the three constructs under study was acknowledged as proposed by Ceja and Navarro [70]. Third, ESM was used to assess situational flow. As previously mentioned, ESM is a powerful method for capturing psychological phenomena as they occur in subjects' daily lives [64]. Fourth, it relies on a self-leadership training program that has proved effective to increase self-leadership skills of employees in a for-profit organization [12]. Finally, the instruments used to measure self-leadership and flow showed good psychometric properties for the sample.

Some limitations should also be considered. This was an observational study with no control arm. Although this precludes us from attributing between-groups differences to the intervention delivered, i.e., self-leadership training in this case, it allows us to assess the temporal relationship of the outcomes of interest [94]. Moreover, the self-leadership training program delivered here followed the one previously developed and implemented by Marques-Quinteiro et al. [12]

concerning the structure and contents and its effectiveness assessment was not a specific goal of this study. A second limitation of our study is the lack of randomization in item presentation during ESM data collection. As such, the construct 'flow' was always activated before 'happiness', which might have induced a response bias to the dataset due to item context (e.g., Christensen et al. [64]). A different approach, to consider in following studies would be to individually assess each of these constructs, at different random moments of the day and/or to randomize the items' presentation. A third limitation is that flow was measured through both dispositional and situational measurements while happiness was measured only in its situational dimension, limiting the study of the association between these two variables. Following studies should consider the two forms of evaluating both variables.

## Conclusions

Theoretical and empirical observation supports a positive association between self-leadership development and subjects' positive affect, with obvious positive outcomes for the organizations themselves. Results herein fit this expectation and go further by showing that the training, development and practice of self-leadership skills contribute to shorten the distance between perceived challenges and skills in job-related tasks, which is a cornerstone condition of flow, as well as to make subjects feel happier in the workplace. As such, the development and practice of self-leadership in the workplace reveals to be a win-win situation with positive outcomes for all involved.

To the best of our knowledge, this is the first longitudinal field study to establish a connection between these three positive psychology related bodies of knowledge: self-leadership, flow, and happiness. By connecting these previously unrelated concepts, this study contributes to the advancement of the theory of positive psychology, while providing preliminary evidence about the usefulness of an organizational tool (self-leadership development programs) for enhancing employees' subjective well-being, namely work-related happiness.

Self-leadership goes beyond self-management (the use of strategies to achieve results externally determined or under external control) and self-regulation (the use of strategies to adapt one's cognition, behavior, motivation, and affect to produce desired outcomes). According to Manz [8] self-leadership enables the individual to set "higher-level control loops" by addressing the superordinate standards for behavior (i.e., its individually chosen reasons). This allows for paradoxical effects such as: enduring externally controlled settings to realize internally motivating purposes. In this sense, self-leadership has the potential to promote self-determination through the increase in autonomy, competence and relatedness. We propose that this can be the mechanism by which self-leadership promotes happiness (through the satisfaction of these three basic psychological needs). It would be highly informative if future research investigates the association between these constructs, by measuring the increase in autonomy, competence and relatedness as mediators of the relationship between self-leadership and happiness.

## Supporting information

**S1 Table. Parameter estimates and 95% confidence intervals (CI) for the effect of between- and within-subject self-leadership on situational flow and happiness data collected using ESM, while controlling for time slot of data collection, sociodemographic and work-related variables.**
(DOCX)

**S2 Table. Variation of self-leadership, situational and dispositional flow, and happiness over the study period provided as estimates for each time of assessment.**
(DOCX)

**S1 Fig. Individual variation for situational flow (red) and happiness (blue) over time collected using ESM.**
(TIFF)

## Acknowledgments

The authors would like to acknowledge Ana Filipa Cruz for her contribution in the online training implementation and data collection.

## Author contributions

**Conceptualization:** Ricardo J. Vargas, Pedro Marques-Quinteiro, Luís Curral.

**Data curation:** Osvaldo Santos, Mónica Fialho.

**Formal analysis:** Ricardo J. Vargas, Osvaldo Santos, Mónica Fialho, Joana Costa, Nicole Eifler, Pedro Marques-Quinteiro, Luís Curral.

**Funding acquisition:** Osvaldo Santos.

**Investigation:** Ricardo J. Vargas, Nicole Eifler.

**Methodology:** Ricardo J. Vargas, Osvaldo Santos, Nicole Eifler, Pedro Marques-Quinteiro, Luís Curral.

**Project administration:** Ricardo J. Vargas.

**Software:** Mónica Fialho.

**Supervision:** Ricardo J. Vargas, Pedro Marques-Quinteiro, Luís Curral.

**Validation:** Pedro Marques-Quinteiro, Luís Curral.

**Visualization:** Mónica Fialho.

**Writing – original draft:** Ricardo J. Vargas, Osvaldo Santos, Mónica Fialho, Joana Costa.

**Writing – review & editing:** Ricardo J. Vargas, Osvaldo Santos, Mónica Fialho, Joana Costa, Nicole Eifler, Pedro Marques-Quinteiro, Luís Curral.

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
