## [Decision Letter · Decision Letter 0]

29 May 2024

PONE-D-24-09967
Lead yourself to the zone and be happy: the effect of self-leadership development on flow and happiness
PLOS ONE

Dear Dr. Vargas,

Thank you for submitting your manuscript to PLOS ONE. After careful consideration, we feel that it has merit but does not fully meet PLOS ONE’s publication criteria as it currently stands. Therefore, we invite you to submit a revised version of the manuscript that addresses the points raised during the review process.

We look forward to receiving your revised manuscript.

Kind regards,

Sitanshu Sekhar Das

Academic Editor

PLOS ONE

“I have read the journal's policy and the authors of this manuscript have the following competing interests: the first and fifth authors are consultants with the organization that delivered the self-leadership training reported in the current paper.”

4. We notice that your supplementary tables are included in the manuscript file. Please remove them and upload them with the file type 'Supporting Information'. Please ensure that each Supporting Information file has a legend listed in the manuscript after the references list.

Additional Editor Comments:

The comments of the reviewers and the points raised are pertinent. I invite you to address all the comments.

Reviewers' comments:

Reviewer's Responses to Questions

**Comments to the Author**

1. Is the manuscript technically sound, and do the data support the conclusions?

Reviewer #1: Partly

Reviewer #2: Partly

2. Has the statistical analysis been performed appropriately and rigorously? 

Reviewer #1: Yes

Reviewer #2: Yes

3. Have the authors made all data underlying the findings in their manuscript fully available?

Reviewer #1: Yes

Reviewer #2: Yes

4. Is the manuscript presented in an intelligible fashion and written in standard English?

Reviewer #1: Yes

Reviewer #2: No

5. Review Comments to the Author

Reviewer #1: Thanks for the given opportunity to review this manuscript. I suggest the following to improve the manuscript standard.

1. At the end of the introduction section add a paragraph to say about this study contribution.

2. In the introduction section I could see the literature review. try to highlight the gap and inconsistencies.

3. Before developing the hypothesis add a paragraph about the theory supporting the theoretical model proposed in this study.

Reviewer #2: Thanks for providing the opportunity to review this paper.

The manuscript examines the relationship between self-leadership, flow and happiness at the workplace. The study also examines the mediating effect of flow on the relationship between self-leadership and happiness.

The study is opportune and relevant in the context of the current unpredictable times, at the individual and the organizational level. The paper is well written and presents fluid thinking. However, I have a few suggestions that the authors may consider:

1. Introduction

I tried looking for the research gap that the authors are trying the bridge. It was difficult to identify the research context in the lines 122-126 that the work is attempting to fulfil. To my mind, this should be brought forward in the structure of the study.

At times, the overall argumentation was in itself answering the question that the authors are trying to respond. For example, in line 164, authors write that “happiness in the 164 workplace is beneficial for both employees and organizations” – which has been empirically proven in the past. In the next lines (165-167), authors argue the benefits of the self-leadership, which should be the part of hypotheses development.

Authors are suggested to rewrite the “Introduction” so as to keep the relevant portions building the overall research context, elucidating the purpose that the research is trying to fulfil. Authors may choose to create a separate heading explaining the variables of the study in detail. For example, lines 58-63 highlight the beneficial effects of self-leadership, followed by the explanation of the mechanism in the next paragraph. The next paragraph (lines 84-94) takes the entire narrative back to the advantages of self-leadership, which takes the readers back to the lines 58-63.

2. Hypotheses under Investigation

I was expecting better and nuanced argumentation here supporting the hypotheses. Authors should highlight the individual hypotheses, building upon the previous literature and their own understanding of the substantive variables of the study. My suggestion is the bring some part of the ‘Introduction’ to this section which potentially may strengthen the overall arguments.

The argumentation for hypotheses 3-4 is based upon a few studies (e.g., Marques-Quinteiro et al., 2019) only. The authors finally ‘predict’ the hypotheses. To my mind, I do not find enough support for the hypotheses 3-4 in the manuscript, and based upon that, author argue in lines 199-201 about the hypotheses, which appears the pretty intuitive. Authors are strongly suggested to strengthen this part.

The overall conceptual model could have been presented to give a better understanding of the hypothesized relationships.

3. Materials and methods

The data was collected in 2016-2020. Though the retrospective analysis was approved by the ethics committee (I do not question that), the entire training was provided following Marques-Quinteiro et al. (2019) (line 233). This chronology is difficult to understand. Authors should explain that how they collected data from a program in 2016, when the training provided was based on a 2019 study.

It is a good practice to provide the in-line citation for methods used (e.g., line 360).

4. Results

Authors should explain the probable reasons behind the non-statistical significance of some of the results (e.g., self-leadership contrast between 3-4). Same goes for happiness also, where half the times, the results were not statistically significant (p-value > 0.05).

5. Discussion

Lines 527-560 are not appropriately placed in the discussion. Line 561 onwards, the findings of the study are discussed. Hypothesis 1 is an important finding of the study, however, I have concerns about the lines 564 – 566 because the study did not examine the ‘strategies’ precisely. The same is reflected in the authors' interpretation of the results also with the word ‘most probably’ (line 564).

Lines 580-593 are avoidable as it takes the readers back to the methodology (e.g., choice of LMM in line 581, line 590).

Authors should discuss the hypotheses individually (e.g., line 594). Hypotheses 2 and 3 are supported together, though in the hypotheses development phase, those were conceptualized differently. Regarding non-supported hypothesis 4, the explanation could be made stronger conceptually, and not just because of the process of data collection. Authors are suggested to look beyond the methods to explain this part.

Overall, this work is of sound relevance to handle the autonomy and self-leadership for the employees working in different organizations. I am hopeful that authors would find the suggestions helpful in developing this study better.

6. PLOS authors have the option to publish the peer review history of their article (what does this mean?). If published, this will include your full peer review and any attached files.

Reviewer #1: **Yes: **Murugan Pattusamy

Reviewer #2: **Yes: **Nishit Kumar Sinha

---

## [Author Response · Author response to Decision Letter 1]

15 Jul 2024

Manuscript number: PONE-D-24-09967

Lisbon, 12 July 2024

Dear Editor-in-Chief of PLOS One, Emily Chenette,

Dear Academic Editor of PLOS One, Sitanshu Sekhar Das

Thank you for the opportunity to submit a revised version of our manuscript, “Lead yourself to the zone and be happy: the effect of self-leadership development on flow and happiness”, for publication in PLOS One.

We appreciate the interest shown in our study and the insightful comments that were made. We have carefully revised the manuscript according to the suggestions of the academic editor and the reviewer, which greatly improved the quality of the manuscript.

Please consider the following answers to the comments/suggestions, one by one. We have also submitted a revised manuscript with changes highlighted, as required.

We look forward to hearing from you soon and we are available to respond to any further questions or comments you may have.

Yours, sincerely,

Ricardo Vargas

Research Center for Psychological Science - CICPSI, Faculdade de Psicologia, Universidade de Lisboa, Lisboa, Portugal

Alameda da Universidade, 1649-013 Lisboa, PORTUGAL

Phone: +351 910787880

E-mail: ricardo.vargas@consulting-house.eu

Note: Lines and page numbers are referred to the version of the manuscript without track changes.

Academic Editor’s Comments to Authors:

Comment: "This does not alter our adherence to PLOS ONE policies on sharing data and materials.”

Academic Editor:

Comment: Thank you for stating the following in the Competing Interests section: “I have read the journal's policy and the authors of this manuscript have the following competing interests: the first and fifth authors are consultants with the organization that delivered the self-leadership training reported in the current paper.”; Please confirm that this does not alter your adherence to all PLOS ONE policies on sharing data and materials, by including the following statement: "This does not alter our adherence to PLOS ONE policies on sharing data and materials.”.

Reply: Thanks for bringing this to our attention. We have now added this information to the Cover Letter. Also, thanks for changing the online submission form on our behalf.

Comment: When completing the data availability statement of the submission form, you indicated that you will make your data available on acceptance. We strongly recommend all authors decide on a data sharing plan before acceptance, as the process can be lengthy and hold up publication timelines.

Reply: We have changed this statement. Our intention is indeed to make data (and statistical scripts) fully available, as indicated in the text: “All collected data are available, together with the R syntax needed to reproduce the results, on Zenodo platform (doi: 10.5281/zenodo.10658188).” (Lines 390-391; Page 17)

Comment: We notice that your supplementary tables are included in the manuscript file.

Reply: Done.

Reviewers' Comments to the Authors:

We would first like to address and acknowledge the positive appreciation of both Reviewers regarding the pre-defined PLOS One Questions. Thanks for the positive appreciations. And thanks also for the constructive and useful additional comments that both Reviewers addressed to us!

Reviewer #1:

Comment: Thanks for the given opportunity to review this manuscript. I suggest the following to improve the manuscript standard.

At the end of the introduction section add a paragraph to say about this study contribution.

Reply: Thanks for the suggestion! We have done so.

Comment: In the introduction section I could see the literature review. try to highlight the gap and inconsistencies.

Reply: Thanks for the suggestion! We have revised the introduction section to make it more explicit about the gaps in the literature that this study now endorses.

Comment: Before developing the hypothesis add a paragraph about the theory supporting the theoretical model proposed in this study.

Reply: Thanks for the suggestion! We have introduced the Job Demands-Resources model as a theoretical framework supporting the definition of the hypotheses (in articulation with self-leadership, flow, and happiness-related theories).

Reviewer #2:

Comment: Thanks for providing the opportunity to review this paper. The manuscript examines the relationship between self-leadership, flow and happiness at the workplace. The study also examines the mediating effect of flow on the relationship between self-leadership and happiness. The study is opportune and relevant in the context of the current unpredictable times, at the individual and the organizational level. The paper is well written and presents fluid thinking.

Reply: We are grateful for these positive regards about the study and manuscript!

Comment: Introduction: I tried looking for the research gap that the authors are trying the bridge. It was difficult to identify the research context in the lines 122-126 that the work is attempting to fulfil. To my mind, this should be brought forward in the structure of the study. At times, the overall argumentation was in itself answering the question that the authors are trying to respond. For example, in line 164, authors write that “happiness in the workplace is beneficial for both employees and organizations” – which has been empirically proven in the past. In the next lines (165-167), authors argue the benefits of the self-leadership, which should be the part of hypotheses development. Authors are suggested to rewrite the “Introduction” so as to keep the relevant portions building the overall research context, elucidating the purpose that the research is trying to fulfil. Authors may choose to create a separate heading explaining the variables of the study in detail. For example, lines 58-63 highlight the beneficial effects of self-leadership, followed by the explanation of the mechanism in the next paragraph. The next paragraph (lines 84-94) takes the entire narrative back to the advantages of self-leadership, which takes the readers back to the lines 58-63.

Reply: Thanks for all these concrete suggestions! They were really helpful in improving the introduction. We have done a thorough (and more complete) revision of the introduction, articulating (with more detailed theoretical support) each of the proposed hypothesis. We believe that the modified introduction points out in a clearer way: (a) the research gaps addressed in this project and (b) avoiding giving the (not correct) idea that the literature already provides enough evidence about our goals and hypotheses. We also rearranged the text so that the reader does not get the idea of repeating or getting back to previous (already explained) ideas/arguments.

Comments: Hypotheses under Investigation: I was expecting better and nuanced argumentation here supporting the hypotheses. Authors should highlight the individual hypotheses, building upon the previous literature and their own understanding of the substantive variables of the study. My suggestion is the bring some part of the ‘Introduction’ to this section which potentially may strengthen the overall arguments.

Reply: We are grateful for this comment! We followed the suggestion, and we have detailed in the introduction section the theoretical backgrounds for each of the hypotheses. We are convinced that the rationale for each of the hypotheses is now much better explained.

Comment: The argumentation for hypotheses 3-4 is based upon a few studies (e.g., Marques-Quinteiro et al., 2019) only. The authors finally ‘predict’ the hypotheses. To my mind, I do not find enough support for the hypotheses 3-4 in the manuscript, and based upon that, author argue in lines 199-201 about the hypotheses, which appears the pretty intuitive. Authors are strongly suggested to strengthen this part.

Reply: Thanks for this call out! We have now explained in more detail the empirical and theoretical considerations that support each of the hypotheses.

Comment: Materials and methods: The data was collected in 2016-2020. Though the retrospective analysis was approved by the ethics committee (I do not question that), the entire training was provided following Marques-Quinteiro et al. (2019) (line 233). This chronology is difficult to understand. Authors should explain that how they collected data from a program in 2016, when the training provided was based on a 2019 study.

Reply: Thanks for raising our attention to this issue, which we understand may be confusing. The fact is that the self-leadership training reported here was developed in 2013, and its first implementation happened in 2014 (before the data collection now reported). The timeline may seem difficult to understand because the results of that first implementation study (in 2014) were published later, in 2018. We have now added more information about this in the manuscript.

Comment: It is a good practice to provide the in-line citation for methods used (e.g., line 360).

Reply: We also agree with this recommendation! We have now added in-line citations supporting the main statistical methods that have been used in the study.

Comment: Results: Authors should explain the probable reasons behind the non-statistical significance of some of the results (e.g., self-leadership contrast between 3-4). Same goes for happiness also, where half the times, the results were not statistically significant (p-value > 0.05).

Reply: Thanks for the suggestion! We have addressed probable reasons for the indicated findings in the discussion section.

Comment: Discussion Lines 527-560 are not appropriately placed in the discussion. Line 561 onwards, the findings of the study are discussed.

Reply: Thanks for this suggestion! We have nevertheless decided to keep part of this content because we believe it is important to briefly describe the evolution of self-leadership competencies evolution (gains/stability) across different intervention (training) moments, and because the sequence of self-leadership specific competences/strategies training was found to be associated with the evolution of flow/happiness.

Comment: Hypothesis 1 is an important finding of the study, however, I have concerns about the lines 564 – 566 because the study did not examine the ‘strategies’ precisely. The same is reflected in the authors' interpretation of the results also with the word ‘most probably’ (line 564).

Reply: Thanks for this highly relevant input! We have revised this part of the text, making it more clear that the found associations are related to the specific strategies that were included in the self-leadership development intervention (avoiding the interpretation of an abstract self-leadership competence improvement).

Comment: Lines 580-593 are avoidable as it takes the readers back to the methodology (e.g., choice of LMM in line 581, line 590).

Reply: We completely agree! We have removed this section from the discussion and introduced part of it in the methods section.

Comment: Authors should discuss the hypotheses individually (e.g., line 594). Hypotheses 2 and 3 are supported together, though in the hypotheses develop-ment phase, those were conceptualized differently. Regarding non-supported hy-pothesis 4, the explanation could be made stronger conceptually, and not just because of the process of data collection. Authors are suggested to look beyond the methods to explain this part.

Reply: Thanks also for this input! We have now discussed the data that support (or not) each hypothesis. We provide possible reasons (besides data collection method) for the non-supported (by data) hypothesis 4.

Comment: Overall, this work is of sound relevance to handle the autonomy and self-leadership for the employees working in different organizations. I am hopeful that authors would find the suggestions helpful in developing this study better.

Reply: Thanks for the positive comments! Reviewers#2’ suggestions were really helpful in improving the manuscript, and we are confident that the new version of the manuscript corresponds to the best expectations.

---

## [Decision Letter · Decision Letter 1]

1 Aug 2025

PONE-D-24-09967R1
Lead yourself to the zone and be happy: the effect of self-leadership development on flow and happiness
PLOS ONE

Dear Dr. Vargas,

Thank you for submitting your manuscript to PLOS ONE. After careful consideration, we feel that it has merit but does not fully meet PLOS ONE’s publication criteria as it currently stands. Therefore, we invite you to submit a revised version of the manuscript that addresses the points raised during the review process.

The reviewers have appreciated the revisions to your manuscript, which address the previous requests. However, a new reviewer has suggested further changes to the Introduction, as well as to the Discussion and Conclusions (see below).

I fully support these suggestions and encourage you to implement them, particularly those regarding the Discussion and Conclusions, which should flow logically from the results and avoid overstatement or speculation.

We look forward to receiving your revised manuscript.

Kind regards,

Francesco Marcatto, Ph.D.

Academic Editor

PLOS ONE

Journal Requirements:

Reviewers' comments:

Reviewer's Responses to Questions

**Comments to the Author**

1. If the authors have adequately addressed your comments raised in a previous round of review and you feel that this manuscript is now acceptable for publication, you may indicate that here to bypass the “Comments to the Author” section, enter your conflict of interest statement in the “Confidential to Editor” section, and submit your "Accept" recommendation.

Reviewer #2: All comments have been addressed

Reviewer #3: (No Response)

2. Is the manuscript technically sound, and do the data support the conclusions?

Reviewer #2: Yes

Reviewer #3: Partly

3. Has the statistical analysis been performed appropriately and rigorously? 

Reviewer #2: Yes

Reviewer #3: Yes

4. Have the authors made all data underlying the findings in their manuscript fully available?

Reviewer #2: Yes

Reviewer #3: Yes

5. Is the manuscript presented in an intelligible fashion and written in standard English?

Reviewer #2: Yes

Reviewer #3: Yes

6. Review Comments to the Author

Reviewer #2: This work has the potential to bring meaningful contribution to the academic literature. I wish you all the best.

Reviewer #3: General comment

The two previous reviewers provided consistent and detailed feedback, primarily addressing theoretical clarity, argumentative structure, and interpretation of results. The authors have responded in a timely and constructive manner, improving the introduction, clarifying time-related aspects, and reorganizing the discussion. While no major issues remain unresolved, some further refinements—particularly in the introduction and discussion—could enhance the overall clarity and consistency of the manuscript. These are outlined in the comments below (Lines numbers are referred to the version of the manuscript with track changes).

Comment on lines 52-63 (introduction):

In the first paragraph, different constructs such as happiness, well-being, and engagement are mentioned without being clearly defined or differentiated. They are presented as if interchangeable, which may confuse readers and weaken the conceptual clarity of the introduction. I recommend clarifying each term and its specific relevance to the study.

Comment on lines 165-166 (introduction):

The sentence “What remains to be known is how this self-regulation process can impact on happiness” could be rephrased for improved clarity and fluency. The expression “what remains to be known” sounds slightly awkward in academic English. You might consider alternatives such as “An open question is...” or “It is still unclear how...”.

Comment on lines 268–271(introduction):

This section seems to contradict earlier parts of the introduction, where job satisfaction is presented as a proxy for happiness. Here, instead, the two constructs are clearly distinguished, with happiness being described as dispositional and less dependent on contextual factors. I would recommend clarifying the relationship between job satisfaction and happiness earlier in the introduction, and maintaining conceptual consistency throughout the section.

Comments on lines 885-893 (Discussion- session “Self-leadership in the organizational context: practical implications for employees’ positive affect):

While this section makes a strong case for the practical value of self-leadership in organizational settings, the argument about productivity may be overstated. The current study did not include a measure of productivity, so any inference about performance outcomes should be made cautiously or supported by external evidence more explicitly.

Comments on lines 946-950 (Conclusions):

This paragraph may slightly overstate the theoretical and practical implications of the findings. While the study provides interesting evidence of associations between self-leadership, flow, and happiness, it does not directly assess general well-being or organizational success. Consider rephrasing to better align claims with the actual data.

7. PLOS authors have the option to publish the peer review history of their article (what does this mean?). If published, this will include your full peer review and any attached files.

Reviewer #2: **Yes: **Nishit Kumar Sinha

Reviewer #3: No

---

## [Author Response · Author response to Decision Letter 2]

14 Aug 2025

Reviewers' Comments to the Authors:

We would first like to address and acknowledge the very positive appreciation of both Reviewers regarding the manuscript.

Note: Lines and page numbers are referred to the version of the manuscript without track changes.

Reviewer #3:

General comment:

The two previous reviewers provided consistent and detailed feedback, primarily addressing theoretical clarity, argumentative structure, and interpretation of results. The authors have responded in a timely and constructive manner, improving the introduction, clarifying time-related aspects, and reorganizing the discussion. While no major issues remain unresolved, some further refinements—particularly in the introduction and discussion—could enhance the overall clarity and consistency of the manuscript. These are outlined in the comments below (Lines numbers are referred to the version of the manuscript with track changes).

Reply: Thank you again for the positive appreciation of our work.

Comment on lines 52-63 (introduction):

In the first paragraph, different constructs such as happiness, well-being, and engagement are mentioned without being clearly defined or differentiated. They are presented as if interchangeable, which may confuse readers and weaken the conceptual clarity of the introduction. I recommend clarifying each term and its specific relevance to the study.

Reply: Thank you for noting this. We agree that the introduction here of the concepts well-being and engagement was misleading. We replaced well-being with subjective well-being (line 60), later in the Introduction we defined it in its relationship with happiness (lines 74-76), and we eliminated the concept of engagement (line 63) to avoid any ambiguity.

Comment on lines 165-166 (introduction):

The sentence “What remains to be known is how this self-regulation process can impact on happiness” could be rephrased for improved clarity and fluency. The expression “what remains to be known” sounds slightly awkward in academic English. You might consider alternatives such as “An open question is...” or “It is still unclear how...”.

Reply: Thank you. We followed your suggestion to change the sentence (line 168).

Comment on lines 268–271(introduction):

This section seems to contradict earlier parts of the introduction, where job satisfaction is presented as a proxy for happiness. Here, instead, the two constructs are clearly distinguished, with happiness being described as dispositional and less dependent on contextual factors. I would recommend clarifying the relationship between job satisfaction and happiness earlier in the introduction, and maintaining conceptual consistency throughout the section.

Reply: Thank you for noting this apparent contradiction. We have now further clarified the distinction between job satisfaction and happiness (lines 271-275).

Comments on lines 885-893 (Discussion- session “Self-leadership in the organizational context: practical implications for employees’ positive affect):

While this section makes a strong case for the practical value of self-leadership in organizational settings, the argument about productivity may be overstated. The current study did not include a measure of productivity, so any inference about performance outcomes should be made cautiously or supported by external evidence more explicitly.

Reply: We agree with the reviewer. We have eliminated from the text any assertion of associations between the results of our study and productivity.

Comments on lines 946-950 (Conclusions):

This paragraph may slightly overstate the theoretical and practical implications of the findings. While the study provides interesting evidence of associations between self-leadership, flow, and happiness, it does not directly assess general well-being or organizational success. Consider rephrasing to better align claims with the actual data.

Reply: Thank you for this remark. We have changed the text accordingly, avoiding overstating the practical implications of the findings.

---

## [Editor Report · Decision Letter 2]

20 Aug 2025

Lead yourself to the zone and be happy: the effect of self-leadership development on flow and happiness

PONE-D-24-09967R2

Dear Dr. Vargas,

We’re pleased to inform you that your manuscript has been judged scientifically suitable for publication and will be formally accepted for publication once it meets all outstanding technical requirements.

Kind regards,

Francesco Marcatto, Ph.D.

Academic Editor

PLOS ONE
---

## [Editor Report · Acceptance letter]

PONE-D-24-09967R2

PLOS ONE

Dear Dr. Vargas,

I'm pleased to inform you that your manuscript has been deemed suitable for publication in PLOS ONE. Congratulations! Your manuscript is now being handed over to our production team.

Kind regards,

on behalf of

Dr. Francesco Marcatto

Academic Editor

PLOS ONE